# Small-Target Detection Algorithm Based on Improved YOLOv11n

**DOI:** 10.3390/s26010071

**Published:** 2025-12-22

**Authors:** Ke Zeng, Wangsheng Yu, Xianxiang Qin, Siyu Long

**Affiliations:** 1Graduate School, Air Force Engineering University, Xi’an 710051, China; taisui1109@163.com (K.Z.);; 2School of Information and Navigation, Air Force Engineering University, Xi’an 710077, China; qxxzhijia@126.com

**Keywords:** YOLOv11n, drone, small-target detection, AFPN, MPDIoU, InnerIoU, SPPF, IDC

## Abstract

**Highlights:**

To address the problems of missed detections and false detections in aerial small-target detection under drone scenarios, this paper comprehensively improves the YOLOv11n model from aspects such as network structure and loss function. It not only adds a small-target detection layer but also adopts AFPN as the neck network. The newly proposed improved modules, C3k2_IDC and SCASPPF, further enhance the model performance. Finally, MPDInnerIoU is presented as the loss function to optimize the regression process.

**What are the main findings?**
First, we add a 160 × 160 resolution detection head with AFPN, replace SPPF with SCASPPF (which highlights small target features and suppresses background clutter), optimize the loss function via MPDIoU-InnerIoU fusion, and enhance C3k2 with IDC (which improves localization accuracy and receptive field). These measures collectively boost performance.Second, on the Visdrone2019 dataset, we find that the improved YOLOv11n achieves 39.256% mAP@0.5, a 6.689% gain over the benchmark.

**What are the implications of the main findings?**
First, it provides a new method for small-target detection in drones, demonstrating that integrating non-adjacent feature fusion, attention mechanisms, expanded receptive fields, and improved loss functions can enhance performance.Second, the algorithm can be directly applied to UAV surveillance, rescue, reconnaissance, and environmental monitoring, reducing missed/false detections.

**Abstract:**

Target detection in UAV aerial photography scenarios faces challenges of small targets and complex backgrounds. Thus, we proposed an improved YOLOv11n small-target detection algorithm. First, a detection head is added to the 160 × 160 resolution feature layer, and non-adjacent layer feature is fused via Asymptotic Feature Pyramid Network (AFPN) to alleviate feature loss caused by downsampling and reduce cross-level feature conflicts. Second, the Spatial Channel Attention SPPF (SCASPPF) module replaces the original Spatial Pyramid Pooling-Fast (SPPF) module to highlight key features and suppress irrelevant ones. Moreover, the loss function is enhanced by fusing MPDIoU and InnerIoU to boost detection accuracy. Finally, Inception Deep Convolution (IDC) is adopted to improve the C3k2 module, expanding the model’s receptive field and enhancing small-target detection performance. Experiments on the Visdrone2019 dataset show that the algorithm achieves 39.256% mAP@0.5, 6.689% higher than 32.567% mAP@0.5 of the benchmark model (YOLOv11n).

## 1. Introduction

Nowadays, deep learning-based target detection technology exhibits excellent performance in feature extraction and high-accuracy detection, which can be roughly divided into two categories: single-stage target detection algorithms and two-stage target detection algorithms. Among them, the two-stage target detection algorithm includes R-CNN [1], Fast R-CNN [2], Faster R-CNN [3], etc. Their core logic is to generate region proposals, followed by performing regression and classification tasks on these proposals. Single-stage target detection algorithms include the YOLO series [4], SSD [5], Retina-Net [6], etc. The core idea is to directly complete the prediction of bounding box and categories without generating region proposals. Owing to the high inference speed of single-stage target detection, they are widely applied in real-time application scenarios.

In recent years, drones have been widely adopted in military and civilian fields due to their low cost and high flexibility. Achieving accurate target detection using drones remains a key challenge. However, detection in UAV aerial images is often confronted with small target sizes, blurred features, and background clutter, which pose significant difficulties for small-target detection. To address these challenges, numerous improved approaches have been proposed. Peng et al. [7] incorporated a contextual semantic enhancement module to enhance the representation ability of multi-scale feature maps and the recognition performance of small targets, but the module still exhibits limitations in reducing the false-positive rate for similar targets. Wang et al. [8] used the global Contextual Transformer (CoT) module and max pooling operation to enhance the extraction of texture information for small targets, thereby improving the detection accuracy. Wang et al. [9] introduced the Efficient Multi-Scale Attention (EMA) module into YOLOv8; this module encodes global information and aggregates pixel-level features. However, it is limited by long-distance dependencies and limited flexibility in parameter sharing. Feng et al. [10] introduced the Spatial-Channel Attention Mechanism (SCAM) into the YOLOv5 model, which improves the model’s focus on small target regions by fusing spatial attention and channel attention, but suffers from high computational complexity and increased overfitting risk. Ahmed Gomaa et al. [11] proposed a real-time method combining detection and tracking features, using Top-hat/Bottom-hat transformations, KLT+K-means, and an efficient association algorithm, to address vehicle occlusion, camera movement, and high computational cost in detecting and tracking moving vehicles in aerial videos; more valuable related work can be found in [12,13,14,15,16,17,18]. Sabina Umirzakova et al. [19] proposed Cotton Multitask Learning (CMTL), a transformer-driven multitask learning framework, which achieves cross-task mutual learning and feature preservation via the Cross-Level Multi-Granular Encoder (CLMGE) and Multitask Self-Distilled Attention Fusion (MSDAF), enabling accurate detection for cotton cultivation scenarios. Overall, these improvements offer valuable insights for the research of UAV aerial photography small-target detection.

YOLOv11n [20], released by the Ultralytics team in September 2024, is a new addition to the YOLO family, achieving significant breakthroughs in architectural design, operational efficiency, and multitask capability. Based on this latest model (YOLOv11n), this paper proposes an improved small-target detection algorithm, with the key modifications as follows: Firstly, a detection head is added to the 160 × 160 feature layer of the backbone network, and the multi-scale feature is fused via the Asymptotic Feature Pyramid Network (AFPN). Secondly, the SPPF module optimized by the Spatial-Channel Attention Mechanism (SCASPPF, as defined earlier) is adopted to replace the original SPPF module. In addition, a new IoU-based loss function, MPDInnerIoU Loss (combining MPDIoU and InnerIoU) is designed to optimize the bounding box regression and improve the detection accuracy. Finally, an improved C3k2 module (named C3k2_IDC) is developed to significantly expand the receptive field with only a slight increase in the number of parameters, thereby enhancing the overall performance of the model.

## 2. Related Work

At present, numerous research approaches for small-target detection tasks have been proposed, and these methods can be applied to the improvement of the YOLO series models, to significantly enhance small-target detection performance. Specifically, data augmentation technologies such as Mosaic data augmentation [21] are used to solve the problems of small-target data volume and uneven distribution. The latest multi-scale feature fusion neck structures such as BiFPN [22] and RCA [23] are adopted to enhance the feature expression ability of small targets. Attention mechanisms such as EMA [24] and SCAM [25] are incorporated into the model to highlight key features of small targets and mitigate background clutter.

The issues of small-target detection tasks (e.g., false detection and missed detection) can be effectively addressed by integrating the aforementioned methods into YOLOv11. Evolved from YOLOv8, YOLOv11 replaces C2f with C3k2, incorporates the C2PSA module into the backbone network, and optimizes the detection head using depthwise separable convolutions. Its architecture is divided into three parts: backbone, neck, and head. The backbone network is built upon Cross Stage Partial Darknet-53 (CSPDarknet-53) [21]. The Conv module in the backbone network first conducts convolution operations, then applies batch normalization, and finally activates the output via the SiLU activation function. Additionally, the backbone network also incorporates the Spatial Pyramid Pooling Fast (SPPF) module, which pools feature maps to a fixed size, thereby increasing the diversity of feature denotation. Finally, a Cross Stage Partial with Pyramid Squeeze Attention (C2PSA) module is incorporated at the end of the backbone network, which integrates the CSP (Cross Stage Partial) structure with the PSA (Pyramid Squeeze Attention) attention mechanism to enhance the feature extraction capability. At the same time, the parametric module C3k2 is designed. (k is an adjustable kernel size, such as 3 × 3 or 5 × 5; 2 indicates using two Bottleneck modules or C3k modules.) It is an improved design based on the traditional CSP Bottleneck with three convolutions (C3) module, and introduced multi-scale convolution kernels C3K. C3k2 can be dynamically adjusted by the C3k parameter (equivalent to C2f when C3k = false and C3k = true, replacing the internal Bottleneck with C3k) to improve model flexibility. The neck network adopts the PAN+FPN (Path Aggregation Network+Feature Pyramid Network) strategy to achieve multi-scale feature fusion.

The detection head employs a decoupled architecture, with separate branches for predicting class probabilities and localization information, and employs task-specific loss functions: Binary Cross-Entropy (BCE) loss for classification tasks, and Distribution Focal Loss (DFL) and Complete-IoU (CIoU) loss for bounding box regression tasks. It also incorporates depthwise separable convolution to reduce the number of parameters and computational complexity, and outputs feature maps at three scales: P3, P4, and P5, which are used for the detection of small, medium, and large targets, respectively. Together, these improvements have contributed to a substantial improvement in the performance of YOLO models.

The model structure of YOLOv11n is illustrated in Figure 1. Compared to prior YOLO series models, it exhibits advantages in both detection accuracy and inference speed, yet it remains primarily suited for the detection of large-scale targets, while its detection accuracy for small targets remains inadequate. To address this limitation, this paper proposes an improved small-target detection algorithm based on YOLOv11n.

## 3. Proposed Algorithm

The feature maps of each detection head of YOLOv11n suffer from feature loss due to successive downsampling operations. Thus, small target features are enriched through the utilization of the higher-resolution feature layer P2. Meanwhile, the neck network adopts FPN+PAN architecture, which only supports feature fusion at adjacent levels despite the significant multi-scale feature fusion effect, while AFPN addresses this limitation by progressively fusing non-adjacent layer feature. Additionally, the features extracted by SPPF are prone to background clutter, so SCASPPF highlights salient features and suppresses background clutter through the improvement of spatial and channel attention mechanisms. In terms of loss function, the training process of small-target detection is optimized by combining the recently proposed MPDIoU and InnerIoU. Finally, the C3k2 module is enhanced using IDC, which expands the receptive field to extract more global information, thereby significantly enhancing the small-target detection performance. The improved model network structure diagram is illustrated in Figure 2.

### 3.1. P2 Small-Target Detection Layer and AFPN

Downsampling continually reduces the resolution of feature maps, with the resolution halved per round, resulting in only a small number of pixels remaining in the image and almost disappearing features, which is easy to cause false detection and missed detection. Therefore, small-target detection accuracy can be significantly improved by incorporating a high-resolution (160 × 160) small-target detection layer (P2) after the second downsampling. In addition, improving the neck network with the AFPN to fuse cross-level features avoids information loss or degradation in multi-level transmission [26], as illustrated in Figure 3.

As can be seen from the figure, during bottom-up feature extraction in the backbone network, the AFPN gradually integrate low-level, high-level, and top-level features, and fuse different scales of feature maps via upsampling, downsampling, and element-by-element addition. During the fusion process, AFPN also incorporates adaptive spatial fusion operations to mitigate conflicts arising from cross-level feature fusion. Since P5 has less detailed information about small target features, high-resolution feature layers can better retain small target features. Therefore, we only progressively fuse the feature from P2, P3, and P4, without significantly increasing the model’s parameters or computational complexity. We first fuse the feature of P3 and P4 to make the semantic information of P4 and P2 closer. Then, features from P2, P3, and P4 are fused to alleviate the problem of excessive semantic differences and realize the gradual fusion of multi-level features. This network architecture has been widely applied in target detection tasks. For instance, Gao et al. [27] incorporated AFPN into YOLOv8 for road target detection, achieving a 1.5% improvement in mAP@0.5. Xu [28] integrated AFPN into YOLOv4 for the detection and recognition of parasite eggs, and its performance was significantly enhanced.

The multi-scale feature fusion process is realized via the ASFF [29] module. ASFF_2 denotes the fusion of two different hierarchical features, while ASFF_3 denotes the fusion of three different hierarchical features. Taking ASFF_2 as an example, the input feature map is first passed through a trainable convolutional layer to achieve weight learning:(1)$$F_{l}=Conv({Input}_{l}),F_{h}=Conv({Input}_{h})$$

Among them, ${Input}_{l}$, ${Input}_{h}$ respectively denote low-level input feature maps and high-level input feature maps. $F_{h}$, $F_{l}$ respectively denote the low-level feature map and high-level feature map after convolution. Conv denotes convolution operation.

Subsequently, different level features are fused, the steps are as follows:

If L = 0, integrating low-level features into high-level features yields(2)$$F=Concat(Upsample(F_{l}),F_{h}),L=0$$

If L = 1, integrating high-level features into low-level features yields(3)$$F=Concat(Downsample(F_{h}),F_{l}),L=1$$

Among them, Upsample and Downsample respectively denote upsampling and downsampling operations, Concat denotes concatenation operations, F denote the fused feature map, and L is used to determine the hierarchical position of the current ASFF module.

Furthermore, fusing high-level and low-level features via the Concat operation and reducing the number of channels to 2 via 1 × 1 point convolution, we obtain spatial weights using the softmax activation function. The output weights can be expressed as follows:(4)$$W_{1},\;W_{2}\;=\;\mathrm{Softmax}(\mathrm{Conv}\;1\;\times \;1\;(F))\;$$

Among them, Conv 1 × 1 denotes point convolution operation, Softmax denotes performing softmax normalization on the elements at each spatial position along the channel dimension, and $W_{1\;},\;W_{2}$ denotes the corresponding output feature weights.

Finally, multiply the weights W with the fused feature to obtain the final fused feature.

If L = 0,(5)$$y=W_{1}\times Upsample(F_{l})+W_{2}\;\times F_{h}\;,\;L=0$$

If L = 1,(6)$$y=W_{1}\times F_{l}+W_{2}\;\times Downsample(F_{h}),\;L=1\;$$

Y denotes the final fused feature.

### 3.2. Improved SPPF

The continuous max pooling of the original SPPF may result in some spatial information loss, and fails to highlight key features or suppress irrelevant features. Therefore, spatial and channel attention SPPF (SCASPPF) is proposed, which incorporates channel attention mechanism into the original SPPF. The channel attention mechanism is applied immediately after the initial convolution. At this stage, the feature maps have not yet suffered spatial information loss from max pooling. These feature maps are then fused with those after max pooling, enabling the module to capture global information while preserving channel and spatial feature details. Finally, the concatenated feature maps are processed via the spatial attention mechanism to further highlight salient features. Among them, the channel attention is similar to the Squeeze and Excitation (SE) Module. It uses global pooling to reduce the dimension to 1×1×C1, followed by a 1×1 convolution and ReLU activation to generate the channel attention weights. This channel attention is then unfolded to match the dimension of the input. Subsequently, the spatial attention aggregates spatial features using average pooling and max pooling along the channel dimension, and concatenates the two generated feature maps. It finally uses a 7×7 convolution and a sigmoid activation function to output a feature map of size 1×H×W, which is then subjected to element-wise multiplication. Therefore, the SCASPPF generates feature maps that enrich fine spatial and channel information, mitigates background clutter, and enhances the overall performance of the model. Its structure is illustrated in Figure 4 and the calculation process is as follows:

Firstly, the input feature map is processed through 1 × 1 point convolution with trainable weights:(7)f=Conv1×1(input)

The input denotes the input feature map, and f denotes the output feature map.

Subsequently, channel attention weighting and max-pooling are, respectively, performed on the output feature map f:(8)$$f_{1}=CA(f),\;f_{2}={maxpool}_{5\times 5}(f)$$

CA refers to the channel attention mechanism, ${maxpool}_{5\times 5}$ refers to a max pooling operation with a 5 × 5 kernel, a stride of 1, and padding of 2, $f_{1}$ stands for the feature map after channel attention weighting, and $f_{2}$ denotes the feature map after max-pooling.

Then, $f_{2}$ undergoes two consecutive max-pooling operations to obtain $f_{3}$ and $f_{4}$:(9)$$f_{3}={maxpool}_{5\times 5}(f),\;f_{4}={maxpool}_{5\times 5}(f_{3})$$

Thereafter, the concatenation operation is employed to fuse $f,f_{1},f_{2},f_{3}$, and $f_{4}$, resulting in the feature map $f_{5}$.(10)$$f_{5}=Concat(f,\;f_{1},\;f_{2},\;f_{3},\;f_{4})$$

Finally, spatial attention weighting is applied to $f_{5}$, followed by processing through another convolutional layer with trainable weights. Here, $f_{final}$ denotes the ultimately obtained feature map, and SA denotes the spatial attention mechanism.(11)$$f_{final}=conv(SA(f_{5}))$$

### 3.3. MPDInnerIoU Loss Function

MPDIoU is a bounding box similarity comparison measure based on minimum point distance, combining multiple factors of IoU, such as overlapping area, center point distance, and size deviation [30]. The calculation process is as follows:

Firstly, calculate the distance $d_{1}$ between the top left corner and the distance $d_{2}$ between the top right corner of the GT box and Prd box.(12)$$d_{1}^{2}={\left(x_{1}^{gt}-x_{1}^{prd}\right)}^{2}+{{(y}_{1}^{gt}-y_{1}^{prd})}^{2}$$(13)$$d_{2}^{2}={\left(x_{2}^{gt}-x_{2}^{prd}\right)}^{2}+{{(y}_{2}^{gt}-y_{2}^{prd})}^{2}$$

Finally, MPDIoU is calculated:(14)$$MPDIoU\;=IoU-\frac{d_{1}^{2}+d_{2}^{2}}{w^{2}+h^{2}}$$

For the ground truth (GT) box, the top-left corner coordinates are ${(x}_{1}^{\mathrm{gt}},{\;y}_{1}^{\mathrm{gt}})$, and the bottom-right corner coordinates are ${(x}_{2}^{\mathrm{gt}},{\;y}_{2}^{\mathrm{gt}})$; for the predicted (Prd) box, the top-left corner coordinates are ${(x}_{1}^{\mathrm{prd}},{\;y}_{1}^{\mathrm{prd}})$, and the bottom-right corner coordinates are ${(x}_{2}^{\mathrm{prd}},{\;y}_{2}^{\mathrm{prd}})$. The variables h  and w respectively denote the height and width of the smallest enclosing box.

InnerIoU optimizes the model training process via auxiliary boxes of varying sizes, with large-sized auxiliary boxes accelerating the convergence speed of small IoU samples and small-sized auxiliary frames accelerating the convergence speed of large IoU samples [31].

As illustrated in Figure 5, the center point of the ground truth (GT) box is ($x_{c}^{\mathrm{gt}}$, $y_{c}^{\mathrm{gt}}$), and the width and height is denoted as $w^{\mathrm{gt}}$, $h^{\mathrm{gt}}$, respectively. Similarly, the center point of the predicted box is ($x_{c\;}$, $y_{c}$), the width and height are denoted by w, h, and R is the scale factor. The calculation method for InnerIoU is as follows:

Firstly, the boundary positions of the auxiliary boxes belonging to the ground truth (GT) box are calculated using the scale factor R, and $x_{l}^{\mathrm{gt}},x_{r}^{\mathrm{gt}},y_{t}^{\mathrm{gt}},y_{b}^{\mathrm{gt}}$ denotes the left boundary x-axis, right boundary x-axis, top boundary y-axis, and bottom boundary y-axis of the auxiliary boxes belonging to the ground truth (GT) box.(15)$$x_{l}^{\mathrm{gt}}=x_{c}^{\mathrm{gt}}-\frac{w^{\mathrm{gt}}\times R}{2},x_{R}^{\mathrm{gt}}=x_{c}^{\mathrm{gt}}+\frac{w^{\mathrm{gt}}\times R}{2}$$(16)$$y_{t}^{\mathrm{gt}}=y_{c}^{\mathrm{gt}}+\frac{h^{\mathrm{gt}}\times R}{2}\;,y_{b}^{\mathrm{gt}}={\;y}_{c}^{\mathrm{gt}}-\frac{h^{\mathrm{gt}}\times R}{2}$$

Use the same method to calculate the boundary positions of the auxiliary boxes belonging to the predicted box. $x_{l},x_{r},y_{t},y_{b}$ denotes the left boundary x-axis, right boundary x-axis, top boundary y-axis, and bottom boundary y-axis of the auxiliary boxes belonging to the predicted box.(17)$$x_{l}=x_{c}-\frac{w\times R}{2},x_{R}=x_{c}+\frac{w\times R}{2}$$(18)$$y_{t}={\;y}_{c}+\frac{h\times R}{2},y_{b}={\;y}_{c}-\frac{h\times R}{2}$$

Calculate the size of the intersection area inter between two auxiliary boxes.(19)$$inter=(min(x_{r}^{gt},x_{r})-max(x_{l}^{gt},x_{l}))\times (min(y_{t}^{gt},y_{t})-max(y_{b}^{gt},y_{b})$$

Calculate the size of the shared area union between two auxiliary boxes.(20)$$\mathrm{union}=(w^{\mathrm{gt}}\;\times \;h^{\mathrm{gt}})\;\times \;R^{2}+(w+h)\;\times \;R^{2}-\mathrm{inter}\;$$

Finally, calculate InnerIoU.(21)$$\mathrm{InnerIoU}=\frac{\mathrm{inter}}{\mathrm{union}}$$

For small-target detection, the sizes of ground truth boxes and predicted boxes are extremely small, rendering it prone to situations where IoU is very small or even zero in the early stages of training. In such cases, the gradient provided by IoU becomes very small or even vanishes. However, InnerIoU can make two non-overlapping bounding boxes overlap by enlarging them. Additionally, even if they remain non-overlapping after enlargement, the $d_{1}^{2}$ and $d_{2}^{2}$ terms in MPDIoU do not rely on IoU for calculation; instead, they compute the distance between the top-left corners and the distance between the bottom-right corners of the boxes. This ensures that the overall IoU loss function does not become excessively small, thereby avoiding the problem of extremely small or vanishing gradients. Therefore, the algorithm in this article combines MPDIoU with InnerIoU in the loss function section, and the calculation method is as follows:(22)$$\mathrm{MPDInnerIoU}\;=\;\mathrm{InnerIoU}-\frac{d_{1}^{2}+d_{2}^{2}}{w^{2}+h^{2}}$$

### 3.4. C3k2_IDC

Large kernel depth separable convolution has a large receptive field and can significantly improve model performance, but its efficiency is low. Although small kernel depthwise separable convolutions are fast, their receptive field which is too small leads to a decrease in model performance. Therefore, IDC (Inception Depthwise Convolution) is inspired by the Inception model and decomposes large kernel deep convolution into multiple parallel branches: small-sized square kernel branch; identity mapping branch; orthogonal nuclear branches. IDC can significantly improve model efficiency through parallel branching and partial channel processing, and orthogonal kernel branches can effectively expand the receptive field, resulting in better performance [32].

This article combines IDC with C3k2 to propose a new feature extraction module C3k2_IDC. It uses Inception-style multi-branch deep convolutions (including 3 × 3, 1 × 11, 11 × 1, and identity branches) embedded between the standard convolutions in the original bottleneck sequential structure. By retaining the residual learning framework and hierarchical feature extraction process of the original module, it effectively expands the receptive field while maintaining strong feature extraction. This not only enhances feature representation capability but also avoids potential structural simplification and gradient disappearance issues that could result from directly replacing with parallel deep convolution branches, thereby more effectively balancing receptive field expansion and model efficiency while preserving YOLO’s lightweight characteristics. C3k2_IDC is illustrated in Figure 6.

The detailed algorithm process of IDC is as follows:

Firstly, the number of channels of the input feature map for each branch is calculated:(23)$$c_{1}=In\_channels-3\times In\_channels\times 0.125\;c_{2},c_{3},c_{4}=In\_channels\times 0.125$$

The $c_{1}$, $c_{2}$, $c_{3}$, and $c_{4}$ denote the number of channels of the input feature maps for the identity mapping branch, the small-size square kernel branch, the 1 × 11 orthogonal branch, and the 11 × 1 orthogonal branch, respectively, and In_channels refers to the number of input channels.

Secondly, the split function is used to divide the input feature map into four parts:(24)$$f_{id},\;f_{wh},\;f_{w},\;f_{h}\;=\;\mathrm{split}(\mathrm{input},\;(c_{1},\;c_{2},\;c_{3},c_{4}))$$

The $f_{id},\;f_{wh},\;f_{w},\;f_{h}$ denote the split feature maps and the input denotes the input feature map.

Then, each split feature map is fed into the corresponding branch:(25)$$f_{wh}^{\prime }=\;\mathrm{DWConv}3\times 3(f_{wh})$$(26)$$f_{w}^{\prime }=\mathrm{DWConv}1\times 11(f_{w})$$(27)$$f_{h}^{\prime }=\mathrm{DWConv}11\times 1(f_{h})$$

***DWConv*3 × 3** denotes a depthwise convolution with a kernel size of 3 × 3 and a padding of 1; ***DWConv*1 × 11** denotes a depthwise convolution with a kernel size of 1 × 11 and a padding of 5 along the width dimension; ***DWConv*11 × 1** denotes a depthwise convolution with a kernel size of 11 × 1 and a padding of 5 along the height dimension. $f_{wh}^{\prime }$, $f_{w}^{\prime }$, and $f_{h}^{\prime }$ denote the output feature maps of each branch, respectively.

Finally, $f_{id}$, $f_{wh}^{\prime }$, $f_{w}^{\prime }$, and $f_{h}^{\prime }$ are concatenated via the concatenation operation, where f denotes the final output feature map of the IDC module(28)$$f=concat(f_{id},f_{wh}^{\prime },f_{w}^{\prime },f_{h}^{\prime })$$

The parameter count and computational complexity of the IDC under fixed channel numbers and large tensor sizes is calculated. We specify both the input and output channel numbers as 256, with a width and height of 80 × 80, respectively. Only 3/8 of the channels are selected and equally allocated to the small square kernel branch and the two orthogonal kernel branches, while the remaining 5/8 are reserved for identity mapping. The results show that the IDC has a parameter count of only 992 and FLOPs are merely 198,400. In contrast, a depthwise separable convolution with a kernel size of 7 × 7 has 12,544 parameters and 313,600 FLOPs. This indicates that the IDC significantly reduces the parameter count while expanding the model’s receptive field.

### 3.5. Algorithm Process

The algorithm divides the input image into several grid cells, each of which undertakes the task of predicting the center point falling on the target in its own area, and outputs a fixed-length vector containing bounding box coordinates and category information. After inputting the images into the improved and trained backbone network and neck network, feature maps of different scales are generated through downsampling process such as convolution and pooling, among which high-resolution feature maps are used for small-target detection, and low-resolution feature maps are used for large-target detection. Finally, the algorithm uses non-maximal suppression (NMS) to process the predicted bounding boxes, and the optimal detection results are screened out and superimposed on the original image. The specific process is illustrated in Figure 7 and Algorithm 1.
**Algorithm 1**: Algorithm of improved target detectionInput: Image dataset D.1: for each image I in D do2:      Divide the image into S × S grids.3:      Extract the feature map m through improved YOLOv11n Network4:      Extract feature vectors v through the detection5:      for each v in I
6:          Calculate the best v and delete the remaining v (NMS)7:          Generate test results R8:      end for9: end forOutput: R

## 4. Experimental Results and Analysis

### 4.1. Dataset and Experimental Environment

The dataset used in this experiment is VisDrone2019, an example of which can be seen in Figure 8. The dataset was compiled and constructed by the AISKYEYE team of the Machine Learning and Data Mining Laboratory of Tianjin University, including 6471 training set images, 548 verification set images, and 1610 test set images, covering a total of 10 types of detection targets. There are 353,550 detection targets in the training set, among which the target size is small and the background environment is complex, which brings great challenges to the detection of small aerial targets.

The COCO dataset considers targets with an area smaller than 32 × 32 as small targets. In the VisDrone2019 training set, there are a total of 353,550 detection targets, with the highest numbers being small-sized pedestrians and vehicles. Statistics show that 212,630 targets have a pixel area smaller than 32 × 32, and 34,827 have a pixel area smaller than 10 × 10, posing significant challenges for UAV aerial target detection. This is illustrated in Figure 9.

The training parameters were 200 epochs, 16 batch sizes, and four workers. Gradient descent (SGD) was used for optimization, and training was stopped 50 times without performance improvement. The initial learning rate (lr0) and final learning rate (lrf) were both 0.01. The details are illustrated in Table 1.

### 4.2. Analysis of Ablation Experiment

As illustrated in Table 2, the mAP@0.5 detected by model ① on the dataset Visdrone2019 reached 32.567%. By adding the small-target detection layer P2, the mAP@0.5 of model ② is increased by 4.439%. The network structure of PAN+FPN is replaced with AFPN to obtain model ③, which fully integrates the features of different scales, and the mAP@0.5 is improved by 1.117%. The model ④ is obtained by introducing the spatial and channel attention spatial pyramid pooling-fast module (SCASPPF), highlighting the important features, and the mAP@0.5 is further improved by 0.124% compared with the model ③. The model ⑤ was obtained by replacing C3k2 with C3k2_IDC, which significantly expanded the receptive field, increasing the mAP@0.5 by 0.573% compared with the model ④, and only slightly increasing the number of parameters. Finally, model ⑤ is further trained with MPDInnerIoU to obtain model ⑥, and the mAP@0.5 is increased by 0.556% again, indicating that it is better than the CIoU used by the benchmark model. In addition, the experimental results of models ⑦ and ⑧ show that the effect of MPDInnerIoU is better than that of InnerIoU. Through a series of improvements to YOLOv11n, the detection accuracy was significantly improved by 6.689% in mAP@0.5.

With the gradual integration of various improved modules, the corresponding number of parameters has modestly grown from 2.59 M to 3.30 M. At the same time, the GFLOPs have surged dramatically from 6.4 to 16.3, and the average inference time per image has also increased from 24.4 ms to 44.5 ms. The core reasons are as follows: the P2 feature layer significantly expands the feature map size, and the AFPN increases the computational overhead of multi-scale feature fusion. Although modules such as SCASPPF and C3k2_IDC contribute to performance improvement, they further accumulate computational costs. Specifically, the integration of the P2 feature layer results in an increase of only 0.11 M in parameters and 5.9 in GFLOPs, with the average inference time per image rising by merely 3.5 ms, yet it achieves the maximum single-module performance gain. Secondly, the addition of AFPN leads to the largest increase in parameters (0.54 M), while GFLOPs only increase by 2.8; due to the substantial growth in parameters, the average inference time per image increases by 8.3 ms, and it delivers the second-largest single-module performance gain, second only to that of the P2 feature layer. Overall, the increase in training cost is within an acceptable range, and significant overall performance improvement is achieved.

The indicators of different models in this experiment are illustrated in Figure 10. The comparison of some test sets between YOLOv11n and the improved YOLOv11n is illustrated in Figure 11.

As can be seen from the confusion matrix in Figure 12, the improved model has higher correct classification rates for most target categories compared with the baseline model. Particularly for “people” (an extremely small-sized target category), its correct classification rate has increased from 0.43 to 0.67. This demonstrates that the proposed model fully extracts the feature of small targets, thus enhancing the performance in small-target detection. Overall, this algorithm can significantly enhance the small-target detection capability without sacrificing the large-target detection capability.

### 4.3. Experimental Analysis of MPDInnerIoU Parameter Settings

According to the experimental results in Figure 13, it can be seen that after the final improved model training, the ratio increased from 1.1 to 1.9 mAP@0.5. It shows an upward downward upward trend, with a ratio of 1.3 and 1.7 mAP@0.5. When reaching the extreme value and ratio = 1.3 overall, within the entire interval, mAP@0.5 reached the maximum value of 39.256%. Therefore, the final model selected the MPDInnerIOU loss function with a ratio of 1.3.

### 4.4. Detection Results of Different Sizes

As illustrated in Figure 14, we calculated the AP values for different size ranges of each target in the Visdrone2019 test set. The experimental results in Figure 3 show that the improved model does not perform well when detecting targets with size < 10 × 10, but it is still better than the baseline model. For 10 × 10 < size < 32 × 32, both mAP@0.5 and mAP@[0.5:0.95] show the most significant improvements, fully demonstrating that the improved model significantly enhances small-target detection performance, while its performance for detecting larger targets remains slightly better than the baseline model.

As illustrated in Figure 15, the AP@0.5 for each target size across different models under ablation experiments is calculated. Due to the small number of targets smaller than 10 × 10 pixels in the VisDrone2019 dataset and the extreme difficulty in detecting such tiny targets, there is a slight improvement in their AP performance, which is mainly attributed to the addition of the P2 feature layer. For targets in the size range of 10 × 10–32 × 32 pixels, the AP values gradually increase with the sequential integration of improved modules, representing the most significant improvement across all size intervals. Targets in UAV aerial imagery exhibit large size variations; although small targets dominate, some large targets also exist. Therefore, the proposed algorithm improves the detection performance of small targets without degrading that of large targets. Additionally, the integration of attention mechanisms and AFPN further enhances the detection performance of medium and large targets, albeit to a lesser extent compared to the improvement achieved for small targets. In summary, the proposed algorithm achieves a significant improvement in small-target detection performance.

### 4.5. Experimental Results of Other Datasets

To further validate the model’s generalization, training was conducted on the Aerial Traffic Image dataset, which is used for road traffic detection. This dataset is from the Kaggle platform and contains 1710 training set images, 558 validation set images, and 440 test set images. There is a slight change in the training parameters, only changing epoch = 200 to epoch = 100.

According to Table 3, the model performs well in identifying different vehicles on the road and shows significant improvement compared to the benchmark model.

The experimental results show that both the baseline model and the improved model have high detection accuracy for large vehicles such as bus, freight, truck, etc. The bold numbers in the figure indicate that the model performs better in detecting that category. However, overall, the improved model has slightly higher detection accuracy for large vehicles than the baseline model. The benchmark model has lower detection accuracy for Car and Motorbike with smaller target sizes, only 73.6% and 20.6%, respectively, indicating that small targets are easy to miss and false detection easily occurs. The improved model achieved detection accuracy of 78.2% and 49.3% for Car and Motorbike, respectively, with an increase of 4.6% and 28.7%. In addition, for large-sized targets such as Trucks and Bus, the improved model is not inferior to the baseline model. This is because we retain the P5 detection head, which is specifically designed for large-target detection. Furthermore, due to the integration of the attention mechanism and more advanced feature fusion, certain categories even achieve certain performance improvements. Therefore, the experimental results demonstrate that the improved algorithm can significantly enhance the ability of small-target detection.

### 4.6. Comparative Experiment

The metrics for each model were obtained through experiments on the Visdrone2019 dataset. As illustrated in Table 4, the proposed algorithm achieves 39.3% of mAP@0.5 with a low parameter of 3.30 M and a GFLOPs of 16.3, which is much higher than other classical algorithms for target detection and surpasses many s-size YOLO model variants in terms of detection performance. In terms of the number of model training parameters and GFLOPs, the improved model is far lower than other comparative models, and only slightly higher than YOLOv10n and the baseline model YOLOv11n. However, its mAP@0.5 is much higher than those of these two models, which meet the requirements of high efficiency and high precision. Finally, we also compared our model with several state-of-the-art variants specialized in small-target detection. Compared with YOLO-FEPA and PC-YOLOn, our model achieves significantly higher mAP@0.5, albeit with a slightly larger number of parameters and higher computational complexity than YOLO-FEPA. When compared with Drone-YOLO, our algorithm outperforms it in both detection accuracy and number of parameters. Overall, the improved model proposed in the algorithm of this paper is more suitable for real-time detection of UAV aerial images.

### 4.7. VisDrone2019 Feature Map Visualization

This article selects some images from the VisDrone2019 test set as input, and outputs feature maps in the previous layer of the P3 detection head of YOLOv11n and the previous layer of the improved YOLOv11n P2 detection head. The results are illustrated in Figure 16.

As illustrated in the experimental results of Figure 16, the improved YOLOv11n achieves a higher resolution of output feature maps compared to the original YOLOv11n, ensuring effective preservation of features for small targets. In contrast, the feature of small targets in the output feature maps of YOLOv11n is only denoted by a small number of pixels. Furthermore, as illustrated in Figure 16a, the original YOLOv11n mistakenly recognizes stone pillars that resemble the shape of cars as real cars, and it can also incorrectly identify vehicles reflected on glass surfaces. These phenomena are attributed to interference from complex environments. However, the improved YOLOv11n effectively suppresses such background clutter, eliminating analogous false detection issues. Finally, Figure 16b reveals that the original YOLOv11n suffers from numerous missed detections, and the feature regions corresponding to individual targets in the feature maps are highly scattered. In contrast, by incorporating the P2 layer and the AFPN neck structure, the improved YOLOv11n achieves more concentrated feature regions for each target, thereby enhancing the overall performance of the model.

### 4.8. Comparative Experiment on the Stability of YOLOv11n and the Improved YOLOv11n

As illustrated in Figure 17, this experiment was conducted to compare the mAP@0.5 metric between YOLOv11n and the improved YOLOv11n through five repeated training runs. The results demonstrate that the performance of the improved model is significantly superior to that of the baseline model: the average mAP@0.5 of the improved YOLOv11n reaches 39.134%, representing an increase of approximately 6.54 percentage points compared with 32.591% of the original YOLOv11n. Meanwhile, both models exhibit low standard deviations, with 0.095% for YOLOv11n and 0.1284% for the improved YOLOv11n, indicating that both models possess excellent training stability. The improved YOLOv11n not only effectively enhances the target detection accuracy but also yields reliable results in repeated experiments, thus demonstrating substantial practical application value.

## 5. Conclusions

Small-target detection in UAV aerial photography scenarios faces challenges such as small target size and a complex background environment. In order to solve these problems, an improved model is proposed based on YOLOv11n: aiming at the problems of insufficient feature and insufficient multi-scale feature fusion of small targets, a small-target detection head and an asymptotic feature pyramid network are introduced. In addition, SCASPPF is used to replace the original SPPF to highlight the salient features of the image and suppress background clutter. At the same time, the integration of MPDIoU and InnerIoU optimizes the training process and significantly improves the detection accuracy. Finally, the C3k2_IDC module is introduced to expand the receptive field of feature extraction and better capture small target features.

Experiments on the VisDrone2019 dataset of the improved model show that the average accuracy of the model is greatly improved compared with the benchmark model, and it is better than most classical target detection models. From the perspective of feature maps, the output feature maps of the improved YOLOv11n are also significantly better than the original model. On the Aerial Traffic Images dataset, the performance improvement is also significant, especially for tiny targets such as motorcycles, where the improvement in detection accuracy is most pronounced.

In general, the improved YOLOv11n algorithm in this paper significantly improves the performance of small-target detection. However, while the improved model achieves significant performance improvements, it is inferior to the baseline model in terms of training speed, model parameters, computational complexity, and single-frame detection speed. Specifically, the number of model parameters has increased by approximately 26%, computational complexity by around 137%, and single-frame detection time by roughly 82%. These drawbacks hinder the model’s practical applications. Therefore, in future work, we will focus on model lightweighting, aiming to slightly reduce model performance while significantly decreasing model parameters and computational complexity.

## Figures and Tables

**Figure 1 sensors-26-00071-f001:**
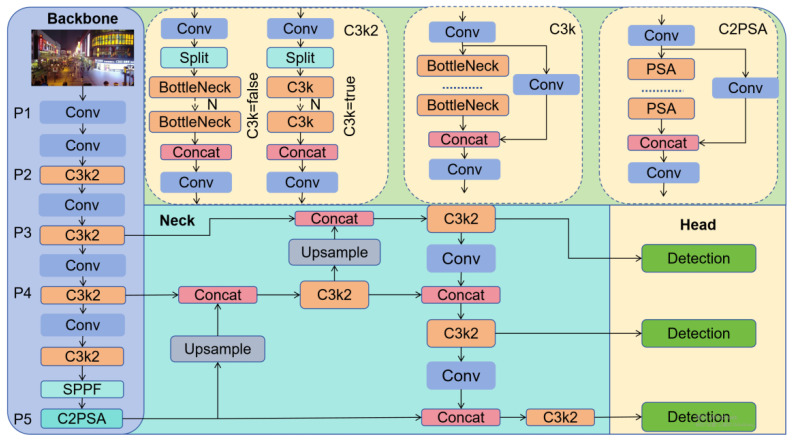
YOLOv11n model structure [20].

**Figure 2 sensors-26-00071-f002:**
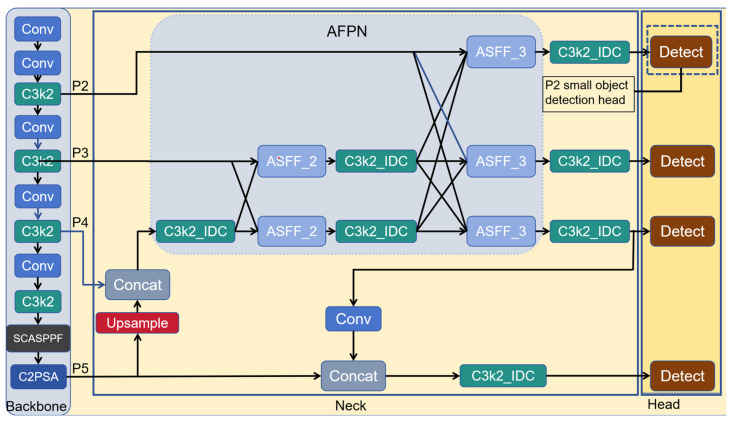
Improved YOLOv11n model. This diagram illustrates the internal structure of the improved YOLOv11n. The AFPN module first fuses features from P2 and P3, then subsequently fuses multi-scale features from P2, P3, and P4. The Spatial Channel Attention SPPF (SCASPPF) module enhances salient features while mitigating background clutter. Finally, all C3k2 modules in the neck network are replaced by C3k2_IDC, thereby expanding the receptive field during feature extraction.

**Figure 3 sensors-26-00071-f003:**
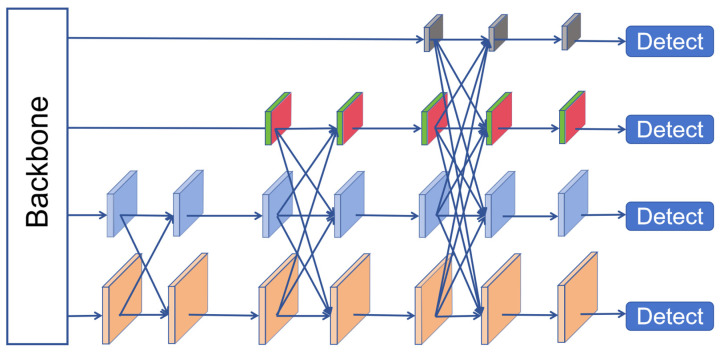
AFPN Structure Diagram.

**Figure 4 sensors-26-00071-f004:**
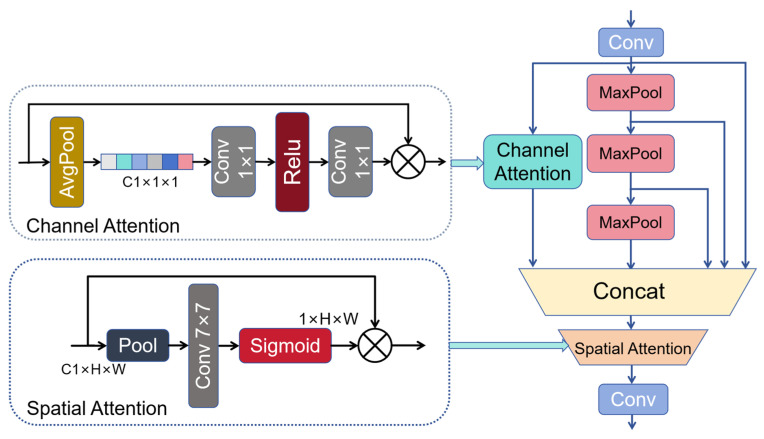
SCASPPF structure diagram.

**Figure 5 sensors-26-00071-f005:**
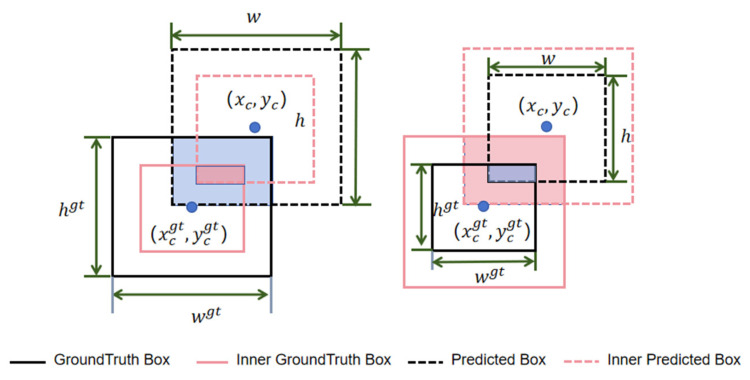
Schematic diagram of InnerIoU.

**Figure 6 sensors-26-00071-f006:**
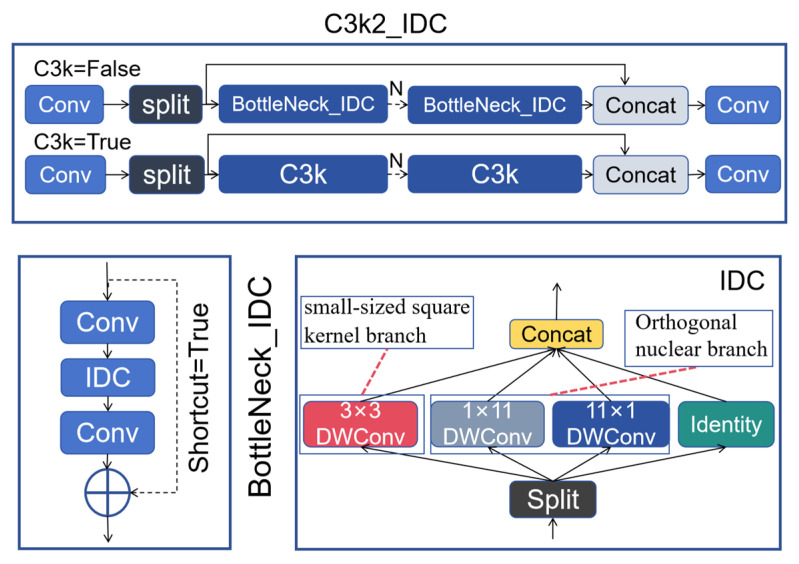
C3k2-IDC Structure Diagram.

**Figure 7 sensors-26-00071-f007:**
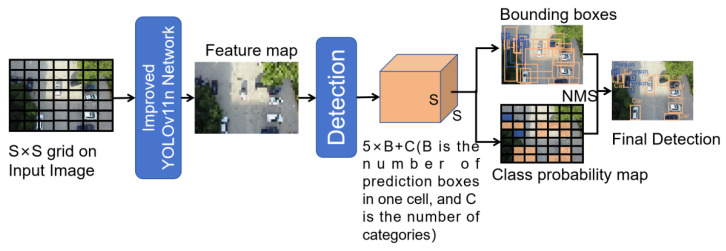
Algorithm process.

**Figure 8 sensors-26-00071-f008:**
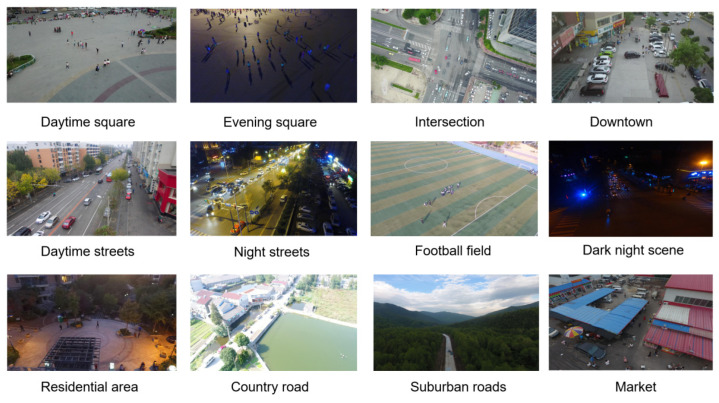
Example of Visdrone2019 dataset.

**Figure 9 sensors-26-00071-f009:**
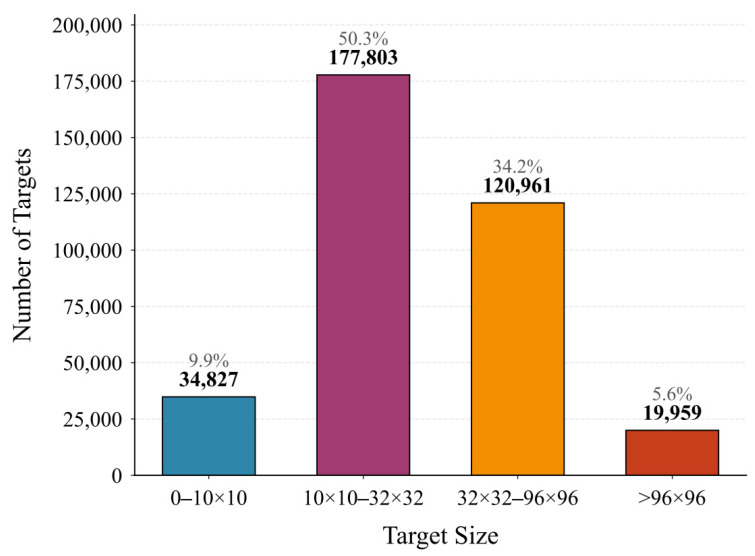
Target pixel area distribution of visdrone2019 training set.

**Figure 10 sensors-26-00071-f010:**
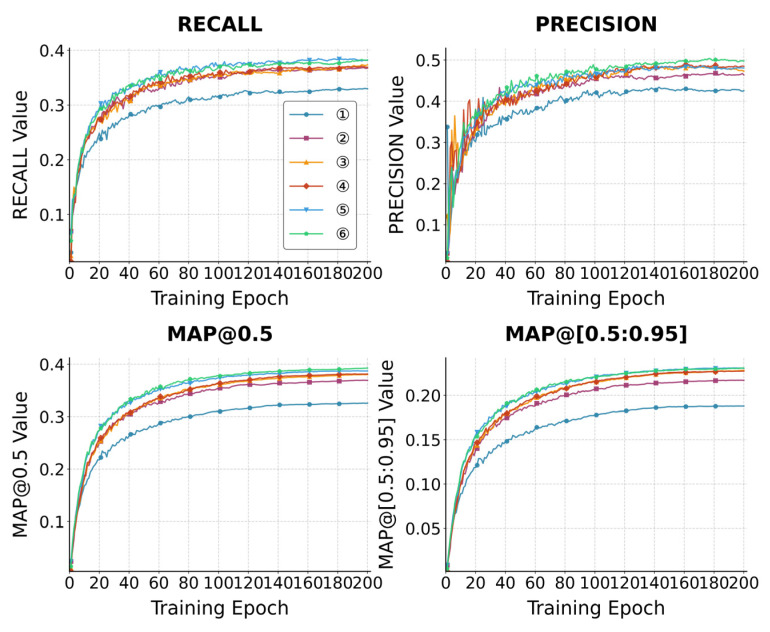
Performance Comparison Results of Different Models.

**Figure 11 sensors-26-00071-f011:**
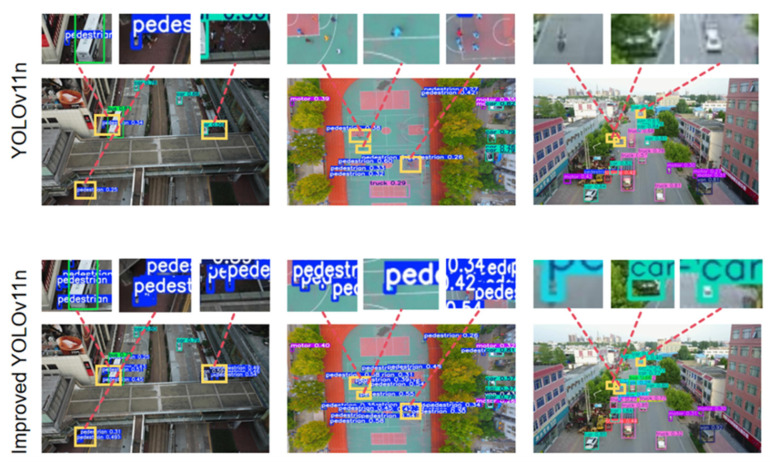
Comparison results of partial test images.

**Figure 12 sensors-26-00071-f012:**
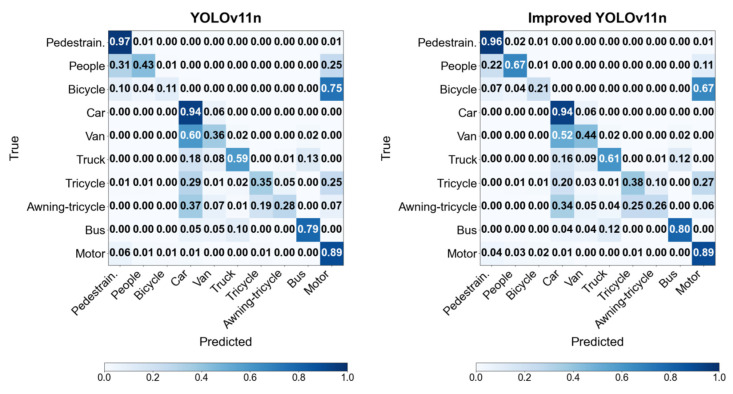
Comparison of YOLOv11n and improved YOLOv11n confusion matrices.

**Figure 13 sensors-26-00071-f013:**
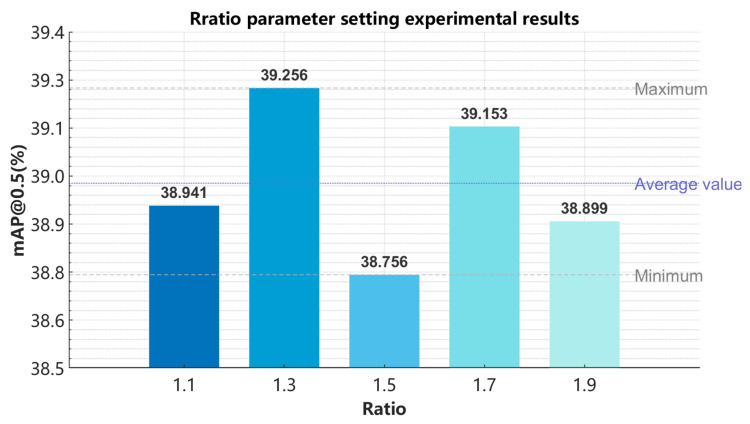
Comparison of MPDInnerIoU with different parameters.

**Figure 14 sensors-26-00071-f014:**
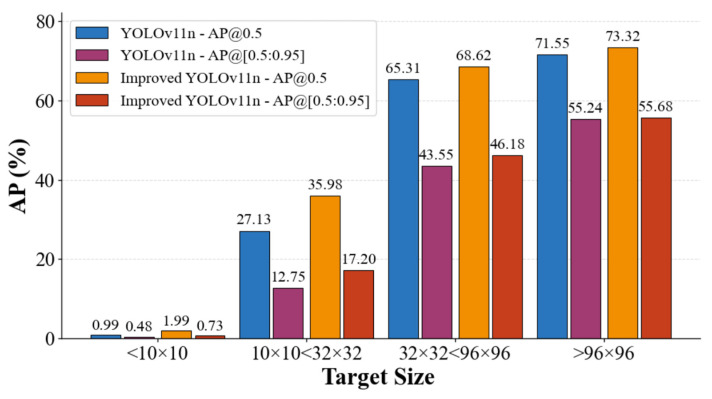
Detection results of different sizes.

**Figure 15 sensors-26-00071-f015:**
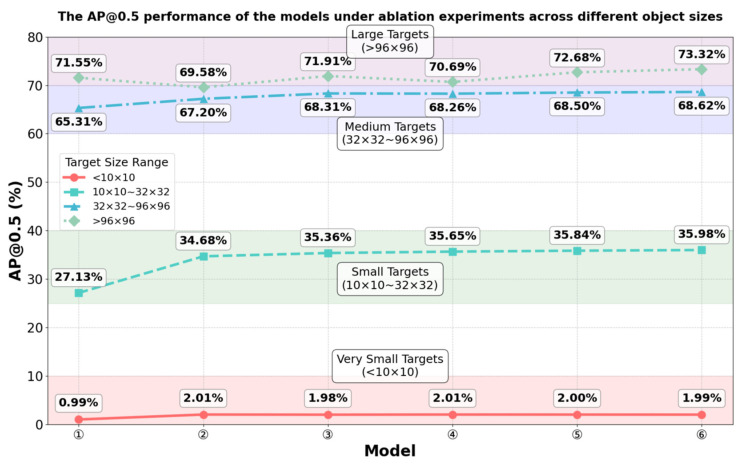
The AP@0.5 performance of the models under ablation experiments across different targets sizes.

**Figure 16 sensors-26-00071-f016:**
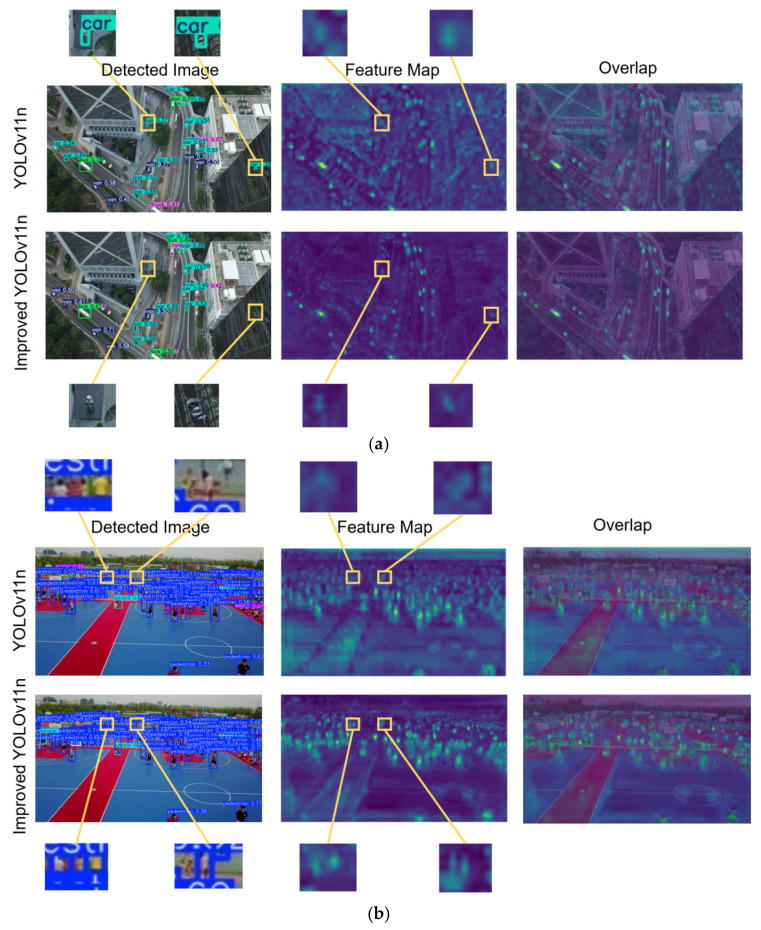
VisDrone2019 feature map visualization. Figure (**a**) illustrates the feature maps under the scenario of false detection, while Figure (**b**) presents those under the scenario of missed detection.

**Figure 17 sensors-26-00071-f017:**
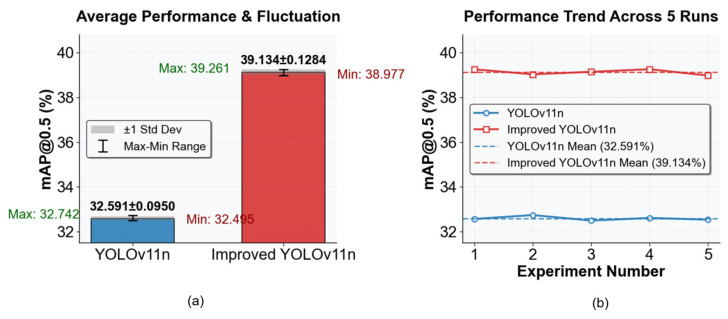
Stability experimental results. Figure (**a**) illustrates the average performance and fluctuations of the improved model and the baseline model; Figure (**b**) depicts the trends of the two models across 5 runs.

**Table 1 sensors-26-00071-t001:** Training Parameters.

Parameter	Parameter Settings
Training Batch (epoch)	200
Batch size	16
Workers	4
Optimizer	SGD
Patience	50
lr0	0.01
lrf	0.01

**Table 2 sensors-26-00071-t002:** Ablation Experiment.

Model	mAP@0.5	Params/M	GFLOPs	Average Inference Time per Image
①: YOLOv11n	32.567%	2.59	6.4	24.4 ms
②: ① + P2	36.916%	2.70	12.3	27.9 ms
③: ② + AFPN	38.033%	3.24	15.1	36.2 ms
④: ③ + SCASPPF	38.157%	3.28	15.2	37.7 ms
⑤: ④ + C3k2_IDC	38.730%	3.30	16.3	43.1 ms
⑥: ⑤ + MPDInnerIoU	39.256%	3.30	16.3	44.5 ms
⑦: ④ + InnerIoU	38.393%	3.28	15.2	37.9 ms
⑧: ④ + MPDInnerIoU	38.537%	3.28	15.2	38.1 ms

**Table 3 sensors-26-00071-t003:** Various detection results of traffic detection dataset.

Category	AP@0.5(YOLOv11n)	AP@0.5 (Improved YOLOv11n)
Articulated-bus	**99.2%**	98.9%
Bus	97.0%	**97.6%**
Car	73.6%	**78.2%**
Freight	**98.3%**	96.8%
Motorbike	20.6%	**49.3%**
Small bus	96.5%	**97.7%**
Truck	75.6%	**84.6%**

**Table 4 sensors-26-00071-t004:** Comparative experiment.

Model	Params/M	mAP@0.5	GFLOPs
Faster-R-CNN	63.20	30.9%	207.0
SSD	12.30	24.0%	63.2
YOLOv5s	9.10	38.8%	23.8
YOLOv8s	11.20	39.0%	28.5
YOLOv11s	9.40	39.0%	21.3
YOLOv10n	2.26	34.2%	6.5
YOLOv10s	7.22	39.0%	21.4
YOLO-FEPA [33]	2.8	36.7%	7.5
Drone-YOLO	3.91	37.0%	-
PC-YOLOn [34]	2.00	36.1%	-
[35]	3.29	38.3%	21.9
Ours	3.30	39.3%	16.3

## Data Availability

The original contributions presented in this study are included in the article. Further inquiries can be directed to the corresponding author(s).

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
