# Peer review of "Small-Target Detection Algorithm Based on Improved YOLOv11n"

_sensors, 2025, doi:10.3390/s26010071_

Round 1

Reviewer 1 Report

Comments and Suggestions for Authors

In the paper “Small object detection algorithm based on improved YOLOv11n” is proposed a small-object-oriented variant of YOLOv11n for UAV aerial imagery. The work presented aims to demonstrate that integrating attention, improved feature fusion, expanded receptive fields, and a hybrid IoU loss improves small object detection performance.

My comments are as follows.

The Abstract should explicitly state the baseline performance, the final performance, and the percentage improvement.

Novelty and contribution clarity: Several improvements, such as P2 layer, AFPN, attention-enhanced SPPF, and IoU-loss variants, have all appeared previously in various YOLO extensions. While the combination is useful, the manuscript needs to emphasize what is genuinely novel versus what is adopted from existing work. Which parts are original contributions? What is the specific innovation in SCASPPF beyond SE-based attention? How is C3k2-IDC different from simply replacing convolutions with InceptionNeXt-style parallel depth wise branches? The current formulation risks appear incremental unless clarified.

The Methodology section needs more mathematical rigor. Some modules are described conceptually but lack: formulas for SCASPPF attention (only qualitative description provided), full algorithmic detail for IDC branches and their shapes, complexity comparisons (parameters, FLOPs per module). To improve reproducibility, authors should add explicit equations or operation-level diagrams.

Regarding MPDInnerIoU Loss, the theoretical justification is insufficient. Although MPDIoU and InnerIoU are individually known, simply combining them via subtraction requires justification. Why does the fusion form (Eq. 16) use a linear combination with fixed weights? Was weighting searched or optimized? How does the loss behave when IoU is small? Is the gradient stable near boundary conditions? Without this, the improvement may be dataset-specific rather than general.

The experimental evaluation requires strengthening.

  • Authors should consider including comparisons with more recent small-object detection baselines, such as: YOLOv9 / YOLOv10 small-object variants, RT-DETR-R / Mobile-DETR lightweight versions, and YOLOF, YOLOS, EfficientViT-Det. Only YOLO and classical detectors are compared; other SOTA small-object detectors are absent.
  • No statistical significance or variation is reported. Results should include multiple runs or standard deviations. A single-run mAP is insufficient for evaluating improvement robustness.
  • Ablation study structure is sound but needs two more analyses: FLOPs and latency per module, and the contribution of each module to small (<10×10), medium, and large object AP (currently only summarized qualitatively).

Some figures are low-resolution or lack clear legends (e.g., Figures 2, 3, 4, 6).

Figure 15 (feature map visualization) should include consistent color maps, normalization, and identical input regions for clarity.

Other comments

The paper contains numerous grammatical errors, inconsistent tences, and unclear sentences. Some examples are: “MPDInnerIoU Loss is introduced to obtain the final improved model…”, “bottle boundary” should be “bottom boundary”, “multi-scale feature information is fused via AFPN to alleviate feature loss caused by downsampling and reduce cross-level feature conflicts”.

Replace all occurrences of “Mosic” with “Mosaic”.

Authors must ensure all symbols in equations are defined at first use.

Table 4: "GDLOPS" should be “GFLOPs”.

Legends in Figures 10–14 are too small and difficult to read.

A thorough English-language revision is necessary.

Reference formatting is inconsistent.

Comments on the Quality of English Language

The manuscript requires substantial improvement in English language quality. Numerous grammatical errors, awkward phrasing, inconsistent terminology, and run-on sentences reduce readability. Several technical descriptions are unclear due to imprecise wording, and some terms are incorrectly translated (e.g., “bottle boundary”). The flow between sentences and paragraphs is often uneven, and figure captions also contain language issues. A thorough English editing or proofreading is strongly recommended.

Reviewer 2 Report

Comments and Suggestions for Authors

This article proposes improvements to the one-shot YOLOv11n model, which specifically aims to enhance the detection of small objects, with a particular focus on the challenges associated with small-object detection in aerial imagery.

Overall, the article is well-structured, with clear sections and images that adequately support the written content. However, throughout most of the manuscript, there are numerous stylistic issues, such as citations (sometimes presented in very small font sizes), formulas (displayed in an unusual and difficult-to-read format), captions of certain figures (e.g., Figure 2), and even some section subtitles (e.g., Section 4). These are formatting problems that the authors are strongly encouraged to address.

Regarding the content, I have the following observations:

- In lines 277–279, the authors write: “From the formula, it can be seen that combining MPDIoU and InnerIoU can accelerate the convergence of the regression process while improving the accuracy of model detection.” As a reader, upon reviewing Formula 16, I do not see why the change from IoU to InnerIoU would accelerate the convergence of the regression process. I strongly suggest that the authors make this point more explicit (preferably in a formal manner).

- In Table 3, is it merely a coincidence that the Baseline model (YOLOv11) achieves better results on classes containing larger objects? Have the authors examined this possibility? If so, it would be interesting to include a few lines explaining their findings.

- The conclusion section provides a summary of the article’s content. However, it does not contribute any new insights for the reader. I suggest that the authors include an analysis of the weaknesses of their proposal (for example, in which cases did the model perform worse than the baseline?). Additionally, what potential future work could be undertaken? (e.g., possible improvements that could be implemented).

Round 2

Reviewer 1 Report

Comments and Suggestions for Authors

Dear Authors,

Thank you for your careful revision.